# OpenReview forum: "Enhancing Safety in Reinforcement Learning with Human Feedback via Rectified Policy Optimization"
_NeurIPS.cc/2025/Conference — NeurIPS 2025 poster_

### Official Review · Reviewer_P8ka · 2025-06-09

**Clarity:** 4
**Significance:** 3
**Originality:** 3
**Rating:** 4
**Confidence:** 4

**Summary:**

The objective of this paper is align large language models (LLMs) to balance helpfulness and safety (i.e., harmlessness). Recent progress on safety alignment, represented as Safe RLHF or SACPO, has been made by solving constrained policy optimization problems under the CMDP framework. The authors identified an issue in the previous works that used expected safety constraints, potentially leading to overly conservative LM policies or safety violations. The authors propose Rectified Policy Optimization (RePO) that defines a problem with critical safety constraints imposed on every prompt. Their experiments show that RePO outperforms strong baseline methods and significantly enhances LLM safety alignment.

**Questions:**

Q1: Could you tell me the reason why Theorem 2 is characterized by $V^r$ and $V^c$, rather than $R$ and $C$? Why does Theorem 2  demonstrates RePO’s ability to guarantee safety while maintaining optimality in the sense of Problem (4)? Perhaps, the authors want to argue that Problem (4) is a special case with $\gamma=1$ and horizon is equal to 1?

Q2: Regarding the following statements in lines 197-199, could you elaborate more? I agree with the following arguments, but I consider that the Slater's condition holds in most cases; thus, strong duality is also usually met. Could you tell me some scenarios where the Slater’s condition does not hold?

> Note that our RePO algorithm is different from the primal-dual methods Dai et al. [7], Huang et al. [16], Wachi et al. [41], which rely on the strong duality of CMDP with the expected constraints [ 27 ]. The property of strong duality is very likely to fail in CMDP with strict constraints where the Slater’s condition does not hold.

**Ethical Concerns:**

["NO or VERY MINOR ethics concerns only"]

**Final Justification:**

I think this is a good paper, but the key ideas are somewhat incremental, as pointed out by other reviewers.

**Limitations:**

- If I understand correctly, there is no discussion regarding limitations, while the authors mentions that "We discuss the future work in conclusion" in line 517.

**Quality:**

3

**Strengths And Weaknesses:**

### Strengths

- This paper is well-written and organized. I found this paper is easy to follow.
- The key ideas behind the problem formulation of RePO are well-explained.
- The empirical evaluation is conducted in a standard manner of safety alignment research.
- The empirical results are very strong. I see that RePO exhibits large improvements compared to SafeRLHF and SACPO.
- The supplementary materials are nice. Please consider releasing the code as well.

### Weaknesses

- While I think Theorem 2 is correct, I am wondering why the authors present bounds with respect to $V^r$ and $V^c$ rather than $R$ and $C$. I consider that it is fairly ok to present Theorem 2, but this theorem does not lead to the theoretical guarantee that the policy obtained by RePO is critically safe.

- It is unclear what is the relation between $R$ and $V^r$ or that between $R$ and $r$.

---

> ### Author Rebuttal · Authors · 2025-07-31
>
> We sincerely thank the reviewer for the encouraging and constructive comments. We will follow your great suggestions to polish our revision (e.g., elaboration on Theorem 2, Slater’s Condition, and limitations). We have attached our code in the supplemental, and we will definitely make it open source. We'd like to address your questions as follows.
>
> **Explanation of Theorem 2.** Thank you for this very insightful comment. The reason we present Theorem 2 w.r.t. value functions as it is more consistent with our implemented algorithm, where it is a token-level MDP formulation with a token-level policy gradient (PPO) update. You are absolutely right, we can consider letting $\gamma=1$ and $H=1$ such that value functions are aligned with the trajectory-level rewards $R(\cdot,\cdot)$ and costs $C(\cdot, \cdot).$ Intuitively, this is more like a bandit-type formulation, and we might need to access the policy gradient w.r.t. the whole trajectory, which is slightly off with our (and current mainstream) implementation for RL finetuning and also may introduce an exponential term w.r.t. the horizon (due to the exponential state and action spaces) [1,2]. We view the value functions as are good approximation to the trajectory-level rewards and costs when $\gamma$ is close to $1$ without introducing exponential dependence in Theorem 2, and we will clarify the difference and add a discussion right after Theorem 2.
>
> **Slater's Condition.** Thank you for raising this great question. Note that our constraint formulation is imposed on every prompt. Let's consider a very hard and unsafe prompt $x$ such that $C(x,y)\geq 0$ for all responses $y$. In other words, you only have to refuse this prompt with $C(x,'\text{refuse}') = 0.$ Therefore, Slater's condition will not hold in this corner case. We do agree that Slater's condition might hold in most settings, and we will include this discussion in our revision.
>
> **Discussions of limitations.** Thank you for your valuable suggestion. We will discuss the limitations in the revised manuscript with an expanded discussion, specifically covering aspects such as the computational overhead inherent to the PPO framework and scalability.
>
> We are truly grateful for your guidance. Please feel free to share any additional feedback, we are glad to have more discussions.
>
> ---
>
> [1] Zhu, Banghua, Michael Jordan, and Jiantao Jiao. "Principled reinforcement learning with human feedback from pairwise or k-wise comparisons." ICML 2023.
>
> [2] Du, Yihan, et al. "Reinforcement Learning with Segment Feedback." ICML 2025.

---

> > ### Comment · Reviewer_P8ka · 2025-08-01
> >
> > Thank you for the authors' feedback.
> >
> > **Slater's condition.** I now understand what the authors mean. Thank you for clarifications.
> >
> > **Explanations of Theorem 2.** I agree with the authors that, when $\gamma = 1$ and $H=1$, the original outcome-level objective functions in (3) are equivalent to the value function with token-level reward and cost functions in Theorem 2. However, in the deviation of Theorem 2, there are many $(1 - \gamma)^{-1}$ and I am not convinced whether Theorem 2 can be directly extended to the theoretical guarantees in terms of the original problem (3). After all, I am not sure what the proposed method guarantees in the sense of the problem (3). I would like to hear some thoughts on this issue from the authors.
> >
> > **Limitations.** If possible, could you provide specific sentences on how the authors will revise the manuscript? I think we can discuss more in this authors-reviewers discussion phase.

---

> > > ### Author Response · Authors · 2025-08-01
> > >
> > > We sincerely appreciate your willingness to engage in further discussions with us and for raising these two interesting questions.
> > >
> > > **Explanations of Theorem 2.**
> > > Thank you very much for your follow-up. We can derive the theoretical performance guarantee for (4) based on Theorem 2 and establish sublinear performance in the order of $\mathcal O(T^{3/4}).$
> > >
> > > Let's focus on discussing the reward and the constraint function can be derived similarly.
> > > Recall the definition of value function $V^r_x(\pi) = \mathbb E[\sum_{h=0}^\infty \gamma^h r(s_h,a_h)|s=x],$ we have $\mathbb E[R(x, y)] - V^r_x(\pi) \leq (1-\gamma^{H})$ for any $x$ and $\pi,$ where $H$ is the maximum generation length.
> > > As discussed above, the value functions are good approximation to the trajectory-level rewards and costs when $\gamma$ or $\gamma^H$ is close to 1. We choose $\gamma ^H = 1 - T^{-\frac{1}{4}}$ and have $$
> > >     (1 - \gamma)^{-1} = (1 - (1 - T^{-\frac{1}{4}})^\frac{1}{H})^{-1}
> > >     \leq (1 - (1-\frac{1}{T^\frac{1}{4}H}))^{-1}
> > >     = T^\frac{1}{4}H,
> > > $$ where the inequality holds because $(1 - T^{-\frac{1}{4}})^\frac{1}{H}\leq 1-\frac{1}{T^\frac{1}{4}H}$ holds according to Bernulli inequality.
> > >
> > > Therefore, we can study the trajectory reward based on Theorem 2 as follows: $$
> > > \begin{aligned}
> > >     \&\sum_{t=0}^T\mathbb E_{\substack{x\sim\mathcal \rho\\\\y^\*\sim\pi^\*(\cdot\mid x)\\\\y\sim\pi^{(t)}(\cdot\mid x)}}[R(x,y^\*)-R(x,y)] \\\\
> > > \&\leq \underbrace{\sum\_{t=0}^T(V\_\rho^r(\pi^\*)-V\_\rho^r(\pi^{(t)}))}\_\text{Regret} + \underbrace{\sum\_{t=0}^T\\left\[|\mathbb E[R(x,y)]-V\_\rho^r(\pi^{(t)})| + |\mathbb E[R(x,y^\*)]-V\_\rho^r(\pi^\*)|\\right\]}\_\text{Approximation Error}\\\\
> > > \&\leq\frac{\lambda_\text{max}\sqrt{(T+1)|\mathcal{A}||\mathcal{S}|\log|\mathcal{A}|}}{(1-\gamma)\sqrt{2}} + \sum_{t=0}^T2T^{-\frac{1}{4}}\\\\
> > > \&\leq\frac{\lambda_\text{max}H(T+1)^\frac{3}{4}\sqrt{|\mathcal{A}||\mathcal{S}|\log|\mathcal{A}|}}{\sqrt{2}} + 2(T+1)^\frac{3}{4},
> > > \end{aligned}
> > > $$ where the first inequality results from the triangle inequality, and the second inequality is obtained by substituting the regret bound in Theorem 2 and the approximation error bound above.
> > > Therefore, the approximation may introduce slightly more regret and violation with  $\mathcal{O}(T^{3/4})$. However, as discussed above, we avoid an exponential term w.r.t. the horizon, which can be regarded as an advantage compared to the bandit-type formulation.
> > >
> > > **Limitations.**  We plan to add a "**Limitations**" section with the following content:
> > >
> > > Although the proposed RePO method achieves significant safety improvements, it has some limitations. First, due to computational resource limitations, we were only able to conduct experiments on the representative 3B and 7B models. Second, similar to PPO, RePO is also an actor-critic style algorithm, requiring additional auxiliary critic models. In RLHF, the critic models usually have a similar scale to the policy model, increasing computational overhead. In the future, we plan to explore efficient LLM safety alignment methods. Fortunately, there are lots of efforts on investigating alternatives to critic models in communities, such as DPO-like methods and GRPO. Combined with our RePO design, they are possible solutions for these challenges in LLM safety alignment.
> > >
> > > Please let us know if you have further questions and comments, we are glad to have more discussions.

---

> > > > ### Comment · Reviewer_P8ka · 2025-08-03
> > > >
> > > > Thank you for the detailed reply. I believe I understand the authors' point now. I would recommend they add some remarks on this in the paper.
> > > >
> > > > I am interested to see how the discussion between the authors and reviewers converges. I will wait to see how the discussion unfolds before making my final evaluation.

---

> > > > > ### Author Response · Authors · 2025-08-03
> > > > >
> > > > > We truly appreciate your time and engagement. We will add a detailed remark as you suggested. We are eager to work with you and the other reviewers to get a positive convergence within the next few days. Please let us know if you have any further comments or concerns, and we would be happy to address them before the discussion period ends.

---

### Official Review · Reviewer_KAzn · 2025-06-30

**Clarity:** 2
**Significance:** 3
**Originality:** 3
**Rating:** 4
**Confidence:** 3

**Summary:**

This work focuses on training LLMs to generate helpful and harmless responses. Building on the SafeRLHF training framework, the authors particularly examine the constrained policy optimization phase and address the issue of safety compensation, which arises from the original CMDP formulation with expected safety constraints. To tackle this, they propose replacing the expected cost-based safety constraints with critical safety constraints, based on a rectified safety metric. Experimental results, compared against state-of-the-art baselines, show that the proposed method, RePO, significantly improves model safety.

**Questions:**

1. The paper states that "$\lambda$ is not a Lagrange multiplier used in the traditional primal-dual method, but rather a non-decreasing rectified penalty." I would appreciate a more detailed clarification of how $\lambda$ is defined in your method, and why it diverges from the classical use of Lagrange multipliers in constrained optimization.

2. Furthermore, how is the value of $\lambda_{max}$ chosen in practice? It would be helpful to understand its influence on the training dynamics and model performance, as well as any sensitivity analysis conducted around this hyperparameter.

3. In Section 3.2, it is unclear whether the cost preference setup follows that of SafeRLHF. The notational reuse suggests that the same structure is adopted, but it is not explicitly stated whether the preferences are defined over cost or reward. Given that SafeRLHF defines preferences in terms of cost, if your method does the same, I strongly encourage you to revise the section to make this distinction clearer. Reusing the same notation for both reward and cost preferences leads to unnecessary confusion.

4. The XS-Test experiments are introduced in the main paper without sufficient context. Readers unfamiliar with this benchmark may struggle to understand the purpose of the experiments and how to interpret the results until they reach the appendix. I recommend including a brief description of XS-Test in the main text to make the paper more self-contained.

5. While the paper checklist claims that the authors have discussed the limitations of the work, no such discussion appears in the main paper. A single vague sentence about future work does not fulfill the requirement for a limitation discussion.

6. Additionally, I was unable to locate the discussion of broader impacts, which is mentioned in the paper checklist that exists in the appendix. In case I missed it, could you please direct me to the section where broader impacts are discussed?

7. Writing and Notation Suggestions:
> (a) Please ensure that all symbols are clearly introduced upon first use, such as, the function $C_\psi(\cdot,\cdot)$ in line 110, as well as symbols like $\eta$, $\alpha$, $\rho$, appear without clear definitions. (b) In line 112, is it a typo as you are referring to a cost function? (c) Is it a typo of $V_\rho^c$ in Theorem 2? (d) The symbols $\phi$ and $\psi$ were used to define both the reward and cost models, as well as the critic networks' parameters. This can lead to confusion and should be resolved through more distinct notation.

**Ethical Concerns:**

["NO or VERY MINOR ethics concerns only"]

**Limitations:**

This work can be viewed as an extension of SafeRLHF aimed at addressing the safety compensation problem. I appreciate the authors' efforts, as well as the theoretical and empirical support provided. However, the paper lacks discussion on how this SafeRLHF framework compares with DPO-like approaches in similarly safety-constrained settings (e.g., SACPO). Both SafeRLHF and RePO require maintaining multiple auxiliary models in addition to the policy model, such as reward and cost models, as well as two critic networks, which significantly increases computational overhead. Despite RePO's superior performance, I believe a discussion of these limitations, along with potential directions for future work, would be a valuable addition to the paper.

**Paper Formatting Concerns:**

I didn't see any major formatting issues in this paper.

**Quality:**

2

**Strengths And Weaknesses:**

**[Strengths]**

This paper addresses the important task of maintaining both helpfulness and harmlessness during LLM fine-tuning. The authors identify and demonstrate a key issue in the SafeRLHF framework, providing a strong motivation for their work. They focus on the constrained policy optimization phase, propose a rectified constraint metric, and support it with both theoretical analysis and a practical algorithm. The effectiveness of the proposed approach is further validated through comprehensive experimental evaluations.

**[Weaknesses]**

Please refer to the Questions and Limitations below.

---

> ### Author Rebuttal · Authors · 2025-07-31
>
> We sincerely acknowledge the reviewers' constructive and positive feedback. We will definitely follow your great suggestions to polish our revision, including modifications to repeated notations, detailed descriptions of XSTest, and identification of typos/imprecise language. Below, we respond to your main questions point by point.
>
> **The rectified penalty design of $\lambda$.** In traditional primal-dual methods, the Lagrange dual multiplier is updated as $\lambda_{t+1} = \lambda_t + \alpha_t \mathbb E[C(x,y)],$ which is used to balance reward maximization and expected safety constraints in the primal optimization. The design may suffer from "safety compensation" due to the expected formulation. In contrast, our rectified penalty is updated as $\lambda_{t+1} = \lambda_t + \alpha_t \mathbb E[\\{C(x,y)\\}^+],$ directly penalizes unsafe via the rectified cost so that the unsafe cost will not be compensated. Note $\lambda_t$ will keep increasing as long as there exist unsafe responses, which guide the policy optimization in $\mathbb E_{x\sim\mathcal{D},y \sim \pi (\cdot\mid x)}[R(x,y) + \lambda_t \\{C(x,y)\\}^+]$ to improve rewards for safe prompts and correct unsafe prompts.
>
> **The choice of $\lambda_{max}$.** We used $\lambda_{max} = 15$ in our experiments as reported in Appendix E. We actually tried various $\lambda_\text{max}$ (e.g., $5, 10$ and $20$) and found that it is quite robust and obtains a similar training performance. We believe the underlying reason may be that the penalty is only imposed on unsafe samples, which becomes small and contributes less penalty to the overall optimization objective during the later training stage. We will add these details in our revision.
>
> **Discussions of limitations and broader impacts.** We acknowledge your great suggestions on the limitations and broader impacts. We will add a more detailed discussion of the limitations and broader impacts in the revision.  Specifically, we will cover computational and memory overhead inherent in the classical PPO finetuning framework and discuss possible solutions like GRPO or DPO-type methods, as you suggested. Thanks again for your insightful suggestion.
>
> We hope that our response addresses the concerns of the reviewer. Please let us know if you have further questions and comments, and we will address them thoroughly.

---

> ### Comment · Reviewer_KAzn · 2025-08-04
>
> Thanks for the responses! After reading the other reviewers' comments and the authors' responses, I will maintain my score.

---

> > ### Author Response · Authors · 2025-08-04
> >
> > Thank you very much again for your great suggestions and for taking the time to engage with us. We sincerely appreciate your acknowledgment and positive feedback on our work!

---

### Official Review · Reviewer_ssaS · 2025-07-02

**Clarity:** 3
**Significance:** 3
**Originality:** 3
**Rating:** 4
**Confidence:** 3

**Summary:**

The paper studies reinforcement learning formulations for safety alignment. It starts with highlighting the limitations of the expected safety constraint adopted in previous work: safe responses can compensate for unsafe responses, leading to suboptimal outcome. The paper then introduces the "critically safe" formulation which imposes a safety constraint on individual prompt-response pairs. The paper then introduced the rectified policy optimization algorithm to solve for the introduced problem. The presented method is shown to outperforms previous formulations in safety.

**Questions:**

Please see weaknesses above.

**Ethical Concerns:**

["NO or VERY MINOR ethics concerns only"]

**Limitations:**

No. The paper needed to discuss the scalability, cost, and stability of the presented algorithm compared to previous work. Also, over-refusal seems to be a major problem of the algorithm.

**Quality:**

3

**Strengths And Weaknesses:**

Strengths:

1. The introduced formulation is sound and well-motivated.

2. The presented algorithm is shown to yield significant empirical results over baselines.

Weaknesses:

1. The presented algorithm results in significantly higher over-refusal rate (20.4% vs. 6.8%), which is concerning by itself. Besides, the paper needs to provide some intuition or (qualitative) analysis to explain that behavior.

2. The paper lacks a discussion of the stability and scalability of the algorithm. It was only applied to a 3B and a 7B model in the experiments.

3. It is not clear why the paper introduces $y_0^{(i)}$ in line 109. It was never used again! Also, line 112 has a typo "can be get".

---

> ### Author Rebuttal · Authors · 2025-07-31
>
> We greatly appreciate the reviewer's constructive comments and positive evaluation of this paper. We'd like to address your questions as follows.
>
> **The cause of over-refusal.** Thank you for your insightful comment. We would like to emphasize that while introducing a 13.60\% over-refusal rate, RePO significantly enhances the safety of LMs (25.50\%) compared to the baseline. This explains that our strict safety formulation and rectified design prioritize safety and induce slightly conservative behavior. We also note that XSTest is a particularly challenging benchmark to test the trade-off between safety and over-refusal. It requires the LLM to have strong abilities to distinguish subtle adversarial prompts, such as *How to steal a car in GTA 4 (a video game)?*. Also, as discussed in [1], *"for many practical applications, it seems reasonable to tolerate some amount of exaggerated safety on safe prompts if this makes models significantly safer on unsafe prompts"*. We believe RePO belongs to this category.
>
> **Stability and scalability.** Thank you for your great comment. We were only able to conduct experiments on the representative 3B and 7B models, mainly due to computational resource limitations. We really want to test our methods in larger models once we have enough resources. We would like to mention that our algorithm is quite stable during training, as in Figure 1, where we run $5$ random seeds and the training curves are stable and similar.
>
> **Clarification on $y_0^{(i)}$.** We apologize for the possible confusion. The notation of $y_0$ is "purely reject" and a reference response to calibrate the cost model such that $C(x,y_0) = 0.$ It is mainly used to illustrate how the cost model is derived. We'll clarify it in revision.
>
> We hope that our response addresses the concerns of the reviewer and is open to further discussion.
>
> ---
>
> [1] Röttger, Paul, et al. "XSTest: A Test Suite for Identifying Exaggerated Safety Behaviours in Large Language Models." NAACL 2024.

---

### Official Review · Reviewer_vTqZ · 2025-07-06

**Clarity:** 3
**Significance:** 2
**Originality:** 1
**Rating:** 3
**Confidence:** 3

**Summary:**

This paper identifies "safety compensation" as an issue where satisfying safety constraints on average allows some LLM responses to be unsafe while others are overly restrictive. To address this, the authors propose Rectified Policy Optimization (RePO), an algorithm that imposes critical safety constraints on every prompt instead of relying on an overall expected value. RePO works by using rectified policy gradients to penalize any strict safety violation, thereby focusing on improving helpfulness only when safety is guaranteed for a given prompt-response pair. Experiments were conducted by fine-tuning Alpaca-7B and Llama3.2-3B models. The results showed that RePO outperformed baseline methods, enhancing LLM safety alignment by increasing the proportion of safe outputs while maintaining helpfulness. For the Alpaca-7B model, RePO achieved a 96.08% safety rate in model-based evaluations and a 90.04% rate in GPT-4 evaluations.

**Questions:**

No.

**Ethical Concerns:**

["NO or VERY MINOR ethics concerns only"]

**Final Justification:**

I am still concerned about the limited applicability of this research. Especially when the formulation and conclusion are closely tied to a single dataset, SafeRLHF. Adding additional results beyond SafeRLHF would significantly reduce my uncertainty about this paper.

**Limitations:**

Although the justification provided by the authors: "We discuss the future work in conclusion." in the Limitation section of the checklist, very little about the method's limitations has been discussed. I kindly suggest that the authors be more upfront about the weakness of the proposed method.

**Paper Formatting Concerns:**

None.

**Quality:**

3

**Strengths And Weaknesses:**

**Strengths:**
- The paper is well-motivated, addressing a clearly defined problem termed "safety compensation". It effectively argues that models can be "expected safe" by satisfying safety constraints on average, yet fail to be "critically safe" by still producing harmful content for individual prompts
- The paper provides a theoretical foundation for its method, presenting theorems that connect the proposed algorithm to the goal of achieving critical safety. The authors support these claims with proofs and provide detailed experimental settings in the appendices, which contributes to the clarity and potential reproducibility of the work.

**Weaknesses:**
- My main concern is the Incremental novelty of this paper. The proposed RePO algorithm could be viewed as an incremental improvement from the paper ([SafeRLHF](https://arxiv.org/pdf/2310.12773)). The core mechanism involves separating the training batch into "safe" and "unsafe" subsets and applying the cost-related penalty term only to the "unsafe" samples. While the paper provides a theoretical rationale for this "rectified" design, the implementation is a modification of the objective function within a framework very similar to that used by its primary baseline, SafeRLHF.
- The paper's central motivation rests on the concept of "safety compensation". However, the evidence supporting the prevalence of this issue is primarily based on a single set of experiments illustrated in Figure 1 and a hypothetical example. The argument could be significantly strengthened by including more diverse quantitative evidence or qualitative analysis, such as examples of actual model outputs, to more concretely demonstrate the problem in practice.
- The paper's empirical validation relies heavily on assets from a single line of prior work, SafeRLHF. Specifically:
  - The fine-tuning process uses prompts from the PKU-SafeRLHF dataset.
  - The reward and cost signals are generated by the beaver-7B-v1.0 models.
  - SafeRLHF is used as the main baseline for comparison.

  While this critique may be beyond the scope of this paper's evaluation, it is worth noting that as a close follow-up to SafeRLHF, the proposed method inherits its predecessor's methodological inflexibility. The reward-cost formulation requires preference pairs for helpfulness and harmlessness to be annotated on the same set of prompts—a criterion that, to my knowledge, perhaps only the BeaverTails and PKU-SafeRLHF datasets currently meet. Consequently, testing the efficacy of the proposed RePO method on other preference datasets is challenging due to this inherited incompatibility with the SafeRLHF formulation. While the authors perform robustness checks using an "unseen internal evaluator" and an "external OOD dataset", the core findings remain situated within a narrow experimental ecosystem, which may not fully reflect the method's generalizability.

---

> ### Author Rebuttal · Authors · 2025-07-31
>
> We are deeply grateful for the reviewer's careful reading and constructive comments. The reviewer’s main concern is the novelty of our algorithm and how it differs from previous methods. We'd like to address your concern in the following.
>
> **Novelty of our algorithm design.** We would like to clarify that RePO has fundamental differences with SafeRLHF w.r.t. problem formulation, algorithm design, and theoretical performance as follows:
> +  **Problem formulation.** SafeRLHF builds on the classical formulation of constrained Markov decision process (CMDP) with the expected safety constraint, i.e., $\mathbb E_{x\sim\mathcal{D}, y\sim\pi_\theta(y\mid x)}\left[C(x,y)\right]\leq 0$. This formulation imposes the expected or average safety compliance and may suffer from "safety compensation" as justified in the paper. To address this issue, we propose a strict safety compliance imposed on every prompt, i.e. $C(x,y)\leq 0,~\forall x\sim\mathcal{D},y\sim\pi_\theta(\cdot\mid x)$.  Unlike the *single* expected constraint in SafeRLHF, we have a potentially large number of per-prompt constraints ($\Omega(|\mathcal D|)$), which are much harder to handle. This motivates our RePO design introduced next.
> + **Algorithm design.** As described above, the single expected safety constraint allows SafeRLHF to adopt the classical PPO-Lagrangian method [1] for finetuning LLMs. This Lagrangian-based approach targets *average* safety by balancing reward and safety through a Lagrange multiplier. It prioritizes optimizing helpfulness (reward) as long as the safety constraint is not violated on average across prompts.
>
>     In contrast, RePO follows a fundamentally different design principle. It targets per-prompt safety by enforcing a rectified constraint on each individual prompt. Specifically, it treats the cumulative constraint violation, denoted by $\lambda$, as a penalty that guides policy optimization. The rectified formulation $\lambda \\{C(x, y)\\}^+$ ensures that safety violations cannot be compensated by safe responses and penalties are only applied to unsafe outputs, not all prompts. This leads to two key advantages:
> 	+ For unsafe prompts, RePO penalizes the LLM to discourage future unsafe generations.
> 	+ For safe prompts, RePO allows the model to focus on improving helpfulness.
>
>   Intuitively, this design is both safety-aware and reward-efficient, as validated by our experimental results in Table 1.
> + **Theoretical performance.** RePO achieves strong performance guarantees in both rewards and safety constraint violations in Theorem 2. To our knowledge, the strict safety guarantee remain largely unexplored in the existing literature and the analysis techniques can be independent of interests.
>
> **The "safety compensation".** We appreciate your suggestion of including more diverse quantitative evidence or qualitative analysis, which would certainly strengthen our argument. We would also like to note that Figure 1 and its complete version of Figure 3 in the Appendix might be comprehensive to show "safety compensation". Figures 1 and 3 show the training curves of SafeRLHF and RePO under three distinct datasets, each containing at least 9K samples. The three datasets were categorized by the initial model's average cost: 1) the initial model is expected to be unsafe with an average cost of 6.682; 2) the initial model is expected to be nearly safe with an average cost of 0.499; 3) the initial model is expected to be safe with an average cost of -3.000. These results demonstrate "safety compensation" of SafeRLHF as follows:
> + Although average cost is under the threshold $0$, $\sim40$ unsafe samples pre-batch remain at the end of SafeRLHF in all three experiments.
> + The curves of average cost and the number of unsafe samples re-increase at $\sim20$th step in the first two column experiments of Figure 3.
>
> We believe these provide strong evidence of the safety compensation effect in SafeRLHF. We will certainly follow your suggestion to evaluate on additional datasets to support our argument further.
>
> **Empirical Validation.** Thank you for your great comment. The main reason for adopting a similar experiment setup with SafeRLHF, is to ensure ***a fair comparison***. The setup in SafeRLHF is standard and widely used in LLM safety alignment studies. We generally follow the same setting and aim to promote reproducible and comparable studies.
> Note we also conducted several additional experiments: 1) fine-tuning and evaluation on Llama3.2-3B; 2) inference on an out-of-distribution dataset from [2]; and 3) evaluation on an over-refusal benchmark. We believe these experiments demonstrate the generality and robustness of our approach.
>
> **The Safety Alignment Paradigm.** Thank you for your insightful suggestion. The current paradigm requires additional (un)safe annotations. It would be very interesting to achieve both helpfulness and safety (or even multiple objectives) with a single type of preference annotation. This is an exciting direction for future research.
>
> **Discussions of limitations.** Thank you for your valuable suggestion. We will discuss the limitations in the revised manuscript with an expanded discussion, specifically covering aspects such as the computational overhead inherent to the PPO framework and scalability.
>
> We hope that our response addresses the reviewer's concerns and that the reviewer can re-evaluate our work. Please let us know if you have any further comments, and we will try our best to address them.
>
> ---
>
> [1] Ray, Alex, Joshua Achiam, and Dario Amodei. "Benchmarking safe exploration in deep reinforcement learning." arXiv:1910.01708. 2019. (OpenAI Safety Gym Project)
>
> [2] Bianchi, Federico, et al. "Safety-Tuned LLaMAs: Lessons From Improving the Safety of Large Language Models that Follow Instructions." ICLR 2024.

---

> > ### Comment · Reviewer_vTqZ · 2025-08-06
> > **Thank you, follow-up question**
> >
> > Thank you to the authors for their thoughtful rebuttal. However, my primary concern regarding the generalizability of the proposed approach remains unaddressed. It is unclear how this method would apply beyond the SafeRLHF dataset, as both the formulation and pipeline appear strongly tied to that specific work. To address this concern, I would recommend providing additional empirical evidence demonstrating that your method is effective on other datasets, or at least outlining your plans for such validation would be appreciated.

---

> > > ### Author Response · Authors · 2025-08-06
> > >
> > > Thank you very much for taking the time to read our rebuttal. We sincerely appreciate your acknowledgment that most concerns have been addressed. We want to answer your follow-up question as follows.
> > >
> > > We would like to clarify that ***the current paradigm requires additional (un)safe annotations/labels for extracting a cost model to justify whether a response is safe or not***. Once we have the cost model, we ***do not*** use any annotated information in the RLHF fine-tuning process except the query prompts. In other words, our method is not limited to any specific dataset as long as we have a good cost model.
> > > We apologize that our previous response might have given the wrong impression that the paradigm needs the (un)safe annotations throughout the RLHF fine-tuning process.
> > >
> > > We greatly appreciate your suggestion to evaluate our method on additional datasets. We plan to extend our experiments using other representative datasets (e.g., HH-RLHF) and alternative cost models for safe RL fine-tuning.
> > > If the reviewer agrees with this plan, we will proceed with the experiments and try to report the results in the revision.
> > >
> > > Lastly, in case you are asking if we can do safe fine-tuning with just a single type of preference annotation without any safe annotations or cost models, we believe this would be a great future direction, where a potential approach is to study how to extract a reliable cost model from the single type of preference annotation.
> > >
> > > We sincerely hope our response addresses your concern and helps make a positive final justification. If you have any further questions or concerns, please feel free to reach out. We will try our best to address them before the discussion ends.

---

> > > > ### Comment · Reviewer_vTqZ · 2025-08-07
> > > > **Thanks**
> > > >
> > > > I don't think fine-tuning with a single preference label is a good idea, nor is it a promising research direction. This is precisely why numerous past works [1,2,3] on preference datasets have advocated for disentangled labels. Additionally, I would recommend moving away from the HH-RLHF dataset and instead using higher-quality datasets like Helpsteer.
> > > >
> > > > That said, I appreciate the authors' thorough rebuttal response. Given their commitment to conducting additional experiments beyond SafeRLHF, I am willing to raise my score.
> > > >
> > > > ---
> > > >
> > > > [1] BeaverTails: Towards Improved Safety Alignment of LLM via a Human-Preference Dataset,
> > > > https://arxiv.org/abs/2307.04657
> > > >
> > > > [2] HelpSteer2: Open-source dataset for training top-performing reward models, https://arxiv.org/abs/2406.08673
> > > >
> > > > [3] RewardBench: Evaluating Reward Models for Language Modeling, https://arxiv.org/abs/2403.13787

---

> > > > > ### Author Response · Authors · 2025-08-07
> > > > >
> > > > > We sincerely thank the reviewer for considering our response and raising the score. Much appreciate! We will proceed with the experiments based on your valuable suggestions. Please let us know if you have any follow-up questions. We will be happy to answer them.

---

### Note · Authors · 2025-08-12

Dear AC and reviewers,

We sincerely thank your time, deep engagement, and insightful suggestions throughout the review and rebuttal processes. Through intensive and constructive discussion, we believe we have addressed the reviewers' concerns, and the major points during the rebuttal process are summarized as follows:

+ Reviewer vTqZ's main concern was the novelty of RePO compared to SafeRLHF. We addressed this by clarifying that RePO has fundamental differences with SafeRLHF from three aspects (i.e., problem formulation, algorithm design, and theoretical performance). We appreciate Reviewer vTqZ's follow-up suggestion on further validating RePO's generalizability with alternative datasets and will try to incorporate them in the revision.

+ Reviewer ssaS's main question was on ``over-refusal''. We answered this by explaining that RePO makes LM significantly safer on unsafe prompts with slightly over-refusal. This is consistent with the discussion and original purpose in the standard over-refusal benchmark [1], which suggests that it is reasonable to tolerate some over-refusal with a large safety enhancement.

+ Reviewer KAzn's questions focused on the rectified penalty design in RePO. We clarified these by
    explaining that it is a non-decreasing penalty-based design on cost functions/unsafe decisions, and
    discussed the detailed difference of the update rules between the rectified penalty and the Lagrange multipliers.

+ Reviewer P8ka's main concern was on the implication of Theorem 2 for the problem formulation (4). We illustrated the key steps to justify the theoretical performance guarantee for (4) based on Theorem 2 with properly chosen discount factors.

We are glad that our responses have resolved these major points
and have been acknowledged by the reviewers. We hope the rebuttal process has achieved a positive convergence.

We truly appreciate the time and effort both AC and the reviewers have invested. We hope this summary provides helpful context for your evaluation.

Best, Authors

----

[1] Röttger, Paul, et al. "XSTest: A Test Suite for Identifying Exaggerated Safety Behaviours in Large Language Models." NAACL 2024.

---

### Decision · Program_Chairs · 2025-09-17

**Decision:**

Accept (poster)

**Comment:**

This paper proposes Rectified Policy Optimization (RePO), which replaces expected safety constraints in SafeRLHF-style formulations with per-prompt “critical safety” constraints, thereby addressing the problem of safety compensation where unsafe responses are averaged out by safe ones. The reviewers agreed that the problem is important and the formulation is well-motivated, with theoretical grounding and strong empirical results showing safety gains over SafeRLHF and SACPO baselines. Strengths highlighted include the clear problem definition, theoretical analysis, and improvements in safety alignment metrics. However, concerns were raised about incremental novelty relative to SafeRLHF, the limited experimental scope (tightly coupled to SafeRLHF datasets and evaluators), and the tradeoff with over-refusal (models refusing safe prompts more often). Reviewers also noted missing discussion of scalability and computational overhead. The authors engaged constructively, clarifying differences from prior work (problem formulation, algorithmic design, theoretical properties), adding explanation of over-refusal behavior, and committing to broader evaluations beyond SafeRLHF. While doubts about generalizability and incremental contributions remain, the rebuttal alleviated some concerns, and reviewers leaned more positive after discussion. Overall, I think it's ok to publish the paper, as long as the authors incorporate all feedback.